# Single-Cell Protein and RNA Expression Analysis of Mononuclear Phagocytes in Intestinal Mucosa and Mesenteric Lymph Nodes of Ulcerative Colitis and Crohn’s Disease Patients

**DOI:** 10.3390/cells9040813

**Published:** 2020-03-27

**Authors:** Laurence Chapuy, Marika Sarfati

**Affiliations:** Immunoregulation Laboratory, CRCHUM, Montreal, QC H2X 0A9, Canada; laurence.chapuy@gmail.com

**Keywords:** mononuclear phagocytes, inflammatory bowel disease, Crohn’s disease, ulcerative colitis, single cell RNA sequencing, flow cytometry

## Abstract

Inflammatory bowel diseases (IBDs), which include Crohn’s disease (CD) and ulcerative colitis (UC), are driven by an abnormal immune response to commensal microbiota in genetically susceptible hosts. In addition to epithelial and stromal cells, innate and adaptive immune systems are both involved in IBD immunopathogenesis. Given the advances driven by single-cell technologies, we here reviewed the immune landscape and function of mononuclear phagocytes in inflamed non-lymphoid and lymphoid tissues of CD and UC patients. Immune cell profiling of IBD tissues using scRNA sequencing combined with multi-color cytometry analysis identifies unique clusters of monocyte-like cells, macrophages, and dendritic cells. These clusters reflect either distinct cell lineages (nature), or distinct or intermediate cell types with identical ontogeny, adapting their phenotype and function to the surrounding milieu (nurture and tissue imprinting). These advanced technologies will provide an unprecedented view of immune cell networks in health and disease, and thus may offer a personalized medicine approach to patients with IBD.

## 1. Mononuclear Phagocytes in Intestinal Mucosa

Genomic, transcriptomic, and proteomic analyses have provided insights into redefining mononuclear phagocyte (MNP) classification and understanding the functional diversity of MNP subsets in tissue at homeostasis and during inflammation. MNPs are now stratified into conventional dendritic cells (cDCs), embryonically-derived macrophages (Mɸ), and monocyte-derived cells (MDC), an entity that regroups monocyte-derived DCs, monocyte-derived Mɸ, and inflammatory monocyte-like cells [1,2,3,4] (Figure 1).

### 1.1. MNPs in Intestinal Mucosa at Steady State

In the gut mucosa, cDCs (cDC1 and cDC2) are relatively conserved between mice and humans [5,6,7,8,9]. Intestinal SIRPα^−^CD103^+^ cDC1 is molecularly related to circulating cross-presenting CD141^+^CLEC9A^+^CADM1^+^ cDC1 and exists as a minor population in the human intestine (5% to 10% of cDCs) [5,8]. cDC2 is classically divided into SIRPα^+^CD103^+^ cDC2, which is the main subset in the small intestine in mice [9] and humans [5,10], and SIRPα^+^CD103^−^ cDC2, which predominates in the colon [5,8] (Figure 2). In mice, at steady state, cDCs ensure tolerance to dietary antigens and enforce a symbiotic relationship with the microbiota [9,10,11,12]. In humans, cDCs isolated from jejunum prime naïve allogenic T cells. However, their ability to polarize T cell responses appears contradictory. Watchmaker et al. showed that SIRPα^−^CD103^+^ cDC1, SIRPα^+^CD103^+^ cDC2 and SIRPα^+^CD103^−^ cDC2 induce Th17, regulatory T cells (Tregs) and Th1 responses, respectively [5]. In contrast, Richter et al. reported that SIRPα^+^CD103^+^ cDC2 promotes Th1 and Th17 responses, SIRPα^+^CD103^−^ cDC2 favours Th1 responses while SIRPα^−^CD103^+^ cDC1 does not induce naive T cell proliferation [13]. Finally, cDCs are more prone to display a regulatory function in distal relative to proximal gut (distal colon versus proximal colon versus ileum) [8,14].

In addition to DCs, all layers of the small and large intestine have a large population of Mɸ (Figure 3). In mice, the main pool of gut Mɸ originates from a constant influx of bone-marrow-derived CCR2^+^ monocytes at steady state [15,16]. Interestingly, colonic and ileal Mɸ express a tissue-specific gene signature using models of adoptive monocyte transfers into Mɸ-depleted recipients [17]. When infiltrating the tissue, recruited monocytes progressively acquire phenotypic and functional characteristics of resident tissue Mɸ. Briefly, CX3CR1^int^Ly6C^hi^CD11c^−^MHCII^−^CD64^low^ monocytes (P1) increase their expression of MHCII, CX3CR1, CD64, and CD11c while decreasing Ly6C expression to gradually progress from CX3CR1^int^Ly6C^+^CD11c^−^MHCII^+^CD64^+^ (P2) to CX3CR1^int^Ly6C^−^CD11c^−/low^MHCII^+^CD64^+^ (P3) cells. Finally, P3 cells differentiate into CX3CR1^hi^Ly6C^−^CD11c^low/+^MHCII^hi^CD64^hi^ mature Mɸ (P4) [15,18]. The latter display potent phagocytic and bactericidal activities and are detected in the lamina propria in close contact with the epithelium, thus favouring interactions with luminal bacteria. Although expressing similar levels of TLRs and CD14 (the co-receptor of LPS) to CX3CR1^int^ Mɸ [15], P4 cells do not produce pro-inflammatory cytokines nor reactive oxygen species under stimulation [3,15], but constitutively secrete IL-10 and low levels of TNF-α and IL-1β. IL-10 produced by CX3CR1^hi^ Mɸ promotes the survival and expansion of FoxP3^+^ regulatory T cells, which are essential for establishing and maintaining oral tolerance [3]. Through chemokine secretion, including the CCR2 ligand CCL2, mature Mɸ participate in the recruitment of their own precursors [19]. Recently, a fraction of this mature Mɸ pool (~35% of Mɸ in P3 and P4 populations) was reported to be maintained locally in the small intestine and colon, independently of monocyte influx [20,21]. These embryonically derived CX3CR1^hi^MHCII^hi^Ly6C^−^TIM-4^+^CD4^+^ Mɸ coexist with TIM-4^−^CD4^+^ and TIM-4^−^CD4^−^ Mɸ that are replenished by circulating monocytes at a slow and fast rate, respectively [20]. TIM-4^+^CD4^+^ Mɸ, deriving first from the yolk sac and next fetal liver progenitors, settle in the deepest layers of the mucosa (sub-mucosa, muscularis externa, and Peyer patches) [21]. scRNAseq analysis further revealed two distinct embryo-Mɸ populations, which both participate in maintaining gut homeostasis. One population cooperates with neurons of the myenteric and submucosal plexus, contributing to intestinal motility, whereas the other subset interacts with blood vessels in the sub-mucosa to ensure vascular integrity [21]. Ginhoux et al. confirmed that gut Mɸ include approximately 20% of embryo-derived Mɸ, which are distinct from the major pool of CD88^+^CD89^+^ monocyte-derived Mɸ, using fate mapping model via Ms4a3-expression history [22].

In humans (Figure 3), a population of anergic HLA-DR^+^ Mɸ , which do not express cell surface markers classically displayed by Mɸ (CD14, CD64, CD32, CD16, CD11b, and CD11c), has been detected in the jejunal mucosa at steady state [23,24]. These cells are highly phagocytic with a potent bacterial killing activity but do not produce inflammatory cytokines in response to LPS stimulation or phagocytosis. Gonzalez-Dominguez et al. showed that 98% of colonic Mɸ are CD14^lo^CD163^+^CD163L1^+^CD209^+^CD11c^−^CD11b^+^ and constitutively produce IL-10 in the mucosa of healthy controls [25]. Other studies reported the presence of CD14^+^ cells in the gut mucosa at steady state. CD14^+^CD33^+^ [26] and CD14^+^HLA-DR^+^ cells [15,27] reside in the ileum of healthy donors. Four Mɸ subsets are settled in the jejunal mucosa, two of them expressing CD11c (CD14^+^HLA-DR^int^ and CD14^dull^HLA-DR^hi^), while the other two CD11c^−^ subsets display a more mature morphology (CD11b^−^CD14^dull^HLA-DR^hi^ and CD11b^+^CD14^hi^HLA-DR^hi^) [28]. Notably, bulk RNA sequencing analysis demonstrated that recently recruited MNPs have a molecular signature closer to circulating monocytes than the three other Mɸ subsets present in the tissue, suggesting a distinct functional nature compared with more mature Mɸ. The mature jejunal Mɸ, like anergic Mɸ, do not secrete IL-10 [24,28].

### 1.2. MNPs in Intestinal Mucosa during Inflammation

Because studying MNPs in human IBD tissue is challenging, MNPs have been largely investigated in mice in T-cell transfer or Dextran Sodium Sulfate (DSS)-induced colitis models. Although no unique experimental animal model of IBD entirely replicates human disease, during colitis, the sequential maturation process of monocytes into anti-inflammatory Mɸ (P4) is interrupted. This arrest of maturation process, in combination with massive recruitment of circulating monocytes, promotes the accumulation of CX3CR1^int^ (P1, P2, and P3) subsets that secrete pro-inflammatory cytokines in inflamed gut [15,18]. Monocytes recruited into inflamed or non-inflamed small intestine were proposed to undergo a distinct maturation program [16]. Several studies provided evidence for similarities between mice and humans regarding MNP nature and function in the gut mucosa. However, some markers broadly used for murine MNP characterization, like Ly6C or CX3CR1, are absent or non-discriminative on monocyte-derived human populations, respectively, limiting the extent of mice–human comparison [29]. CD11c, whose expression level progressively increases on murine MNPs during monocyte maturation cascade [15,18], is elevated in recently recruited human intestinal Mɸ at homeostasis [28] and remains stable during inflammation [15,25,30,31].

CD14, a hallmark of monocytes, is expressed at each step of monocyte differentiation in the murine and human intestine [15,19]. CD14^+^ cells detected in the mucosa of IBD patients were suggested to originate from circulating monocytes [26]. A proof of concept was provided by experiments conducted in 1995 by Grimm et al. showing that radiolabelled-circulating monocytes, isolated from IBD patients and reinfused into the same individual, were retraced as CD14^+^ cells in the inflammatory mucosa [32]. CD14^+^ cells accumulating in the mucosa of patients with CD expressed CD68 [26] and CD68^+^INOS^+^ cells accumulating massively in the subepithelial areas in CD and UC are thought to damage the intestinal barrier by deregulating the cell junction proteins and inducing the apoptosis of epithelial cells [33,34]. Several studies confirmed the presence of CD14^+^ MNPs in the inflamed mucosa of CD and UC patients, both in the small intestine and colon [15,25,26,27,35,36,37,38]. CD14^+^ MNPs include HLA-DR^high^CD163^+^ Mɸ, HLA-DR^dim^ MNPs, HLA-DR^+^CD64^+^ cells, and HLA-DR^+^SIRPα^+^ cells. The latter, which comprises more than 95% of CD14^+^ cells, was shown to accumulate in the inflamed colon in large cohorts of CD and UC patients compared to paired non-inflamed colon, healed mucosa of IBD patients in endoscopic remission, healthy colon of control donors, or inflamed colons of non-IBD patients [31,39]. Stratification of colonic CD14^+^ MNPs according to CD64 and CD163 expression allowed the discrimination of two phenotypically, morphologically, and functionally distinct populations in IBD patients. Only CD14^+^CD64^+^CD163^−/dim^, and more particularly CD163^−^ but not CD163^+^ cells, accumulated in inflamed mucosa of IBD patients in proportions that correlate with endoscopic disease severity in CD, regardless treatment history, demographics and disease behavior or location [31,39]. CD14^+^CD64^+^CD163^−^ MNPs displayed monocyte morphology, whereas CD163^+^ MNPs resembled Mɸ with cytoplasmic vacuoles [31]. In CD and UC patients, both CD163^−^ and CD163^+^ MNPs produced pro-inflammatory cytokines (IL-1β, IL-6, IL-12p40, and IL-23). CD163^+^ MNPs expressed more TNF-α and IL-10 than CD163^−^ MNPs, which are the major contributors to IL-23 and IL1β [31,39]. The CD163^−^, but not CD163^+^, MNPs promoted Th17 and Th17/Th1 memory responses in an IL-1β-dependent manner [31,39], which have been suggested to be pathogenic in IBD [40,41,42,43].

Unsupervised and unbiased scRNAseq analysis, using the Smart-seq2 protocol, which captured the entire HLA-DR^+^SIRPα^+^ population in inflamed colonic mucosa of three CD patients, identified six distinct clusters, two of which were significantly enriched in CD14^+^CD163^−^ and CD14^hi^CD163^hi^ cells. One CD14^+^ cluster that expressed a *TREM1/FCAR/FPR1/S100A9 /C5AR1/SLC11A1/CD300E* gene signature was enriched in CD163^−^ cells, whereas the second one expressing *CD209/MERTK/MRC1/CD163L1* was enriched in CD163^hi^ Mɸ [31]. The four remaining clusters were enriched in cells bearing the gene signature of pDCs (e.g., *TCF4, IL3R,* and *IRF8*), DC precursors (e.g., *SPIB* and *LTB*), cDC2 (e.g., *CD1c, CLEC10A,* and *SLC38A1*) and uncharacterized CD14^dim^ cells, which express myeloid markers (e.g., *CCD88B* and *NRP12*). As expected, conventional HLA-DR^+^SIRPα^−^ cDC1s were not captured in this analysis. Other authors examined an unfractionated LPMC population in the ileal mucosa of 11 CD patients using 10 genomic scRNAseq and high dimensional protein (CyTOF) profiling and stratified the Mɸ population into two subsets: inflammatory and resident Mɸ [34]. The inflammatory Mɸ accumulate in inflamed ileal mucosa of CD patients. Zooming at the latter allowed the segregation of two subsets, defined as program 1 and 2, with program 1 cells displaying a gene signature (*S100A9*/*S100A8*/*VCAN*/*TIMP1*/*C5AR1/IL1b*) close to that of circulating monocytes [44] and colonic CD14^+^CD163^−^ MNPs reported in CD [31]. In UC patients, scRNAseq analysis of LPMC extracted from inflamed mucosa indicated the presence of CD14^+^ Mɸ and CD14^+^ inflammatory monocytes (*FPR1/S100A12/SLC11A1/CD300E)*, with higher frequency of the latter in the inflamed relative to non-inflamed colon [45]. In that regard, unsupervised phenotypic analysis of colonic CD14^+^ MNPs in UC patients was used to discriminate CD14^+^CD163^−^ from CD14^+^CD163^+^ subsets that were best characterized by TREM-1, CCR2, CD11b with or without CLEC5A expression, and CD163, MERTK, CD209, CD206, and CD16, respectively [39]. Finally, mucosal profiling of pediatric-onset IBD further revealed two CD68^+^CD14^+^CD64^hi^ Mɸ subsets with S100A8^+^IL-1β^+^TNFα^+/−^ cells sharing their gene signature (*TREM1*/*S100A9*/*S100A8/*
*SLC11A1/CD300E/FPR1/VCAN)* [46] with inflammatory CD14^+^CD163^−^ MNPs in adult IBD [31].

Collectively, inflamed IBD mucosa is predominantly infiltrated by a swarm of pro-inflammatory CD14^+^CD163^−^ MNPs that cohabit with CD14^+^CD163^+^ Mɸ and cDCs and potentially drive T cell intestinal inflammation in IBD (Figure 4 and Table 1).

### 1.3. CD163^−^ MNPs in Inflamed Intestinal Mucosa: which Monocyte-Derived Cell Types?

Although a consensus appears to have been reached regarding the monocyte origin of intestinal pro-inflammatory CD14^+^ MNPs, debate is ongoing about the monocyte-derived cell type to which CD14^+^CD163^−^ MNPs belong. Furthermore, the identification of the murine counterpart of intestinal pro-inflammatory CD14^+^CD163^−^ MNPs remains unclear.

Single-cell phenotypic and transcriptomic profile of intestinal MNPs suggests that human CD14^+^CD163^−^ cells are related to the murine CX3CR1^int^ subset. Both murine CX3CR1^int^ and human CD14^+^CD163^−^ MNP subsets express high levels of *TREM-1* and *FCAR* (encoding CD89) [31,47]. During inflammation, in DSS-induced and T cell-mediated colitis, CX3CR1^int^ cells are either defined as inflammatory Mɸ [15,18], which are sessile cells unable to migrate, Mo-DCs, or even cDCs, capable of migration and antigen presentation [9,48,49].

The classification of CD14^+^CD163^−^ MNPs infiltrating CD and UC colon into inflammatory monocyte-derived-DCs (Inf Mo-DCs), monocyte-derived Mɸ (Inf Mɸ), monocyte-like cells (Inf Mo-like), or DCs (Inf DCs) remains challenging. Inf Mo-DCs have been described in skin, synovial fluid of patients with rheumatoid arthritis, and tumor ascites [50,51]. The latter are CD14^+/dim^ cells, best characterized by the *CD1c*/*IRF4/FcER1/ZBTB46/CCR7* gene signature; they secrete pro-inflammatory cytokines, augment memory Th cell responses and favour naïve T cell polarization [51]. However, three recent separate studies, using scRNAseq, defined human “CD14^+/dim^ DCs”, a cell type that belongs to CD1c^+^ cDC2 subsets and thus distinct from CD14^+^CD88^+^CD89^+^ monocytes. Firstly, Villani et al. described two distinct cDC2 subsets in the blood of healthy subjects: DC2 (CD14^−^FcεR^+^CLEC10A^+^CD1c^+^ cells) and DC3 (CD14^dim^CD163^+^CD36^+^S100A8^+^S100A9^+^CLEC10A^+^ cells) [52]. Secondly, Dutertre et al. further subdivided DC3 into three subsets: CD14^-^CD163^−^, CD14^-^CD163^+^, and CD14^+^CD163^+^ cells. The circulating CD14^+^CD163^+^ cells represent the Inf DCs, whose proportion is correlated with disease activity index in SLE patients [44]. Brown et al. identified two murine cDC2 subsets in spleen: pro-inflammatory RORγt^+^CLEC10A^+^CLEC12A^+^ “cDC2B” resembling circulating DC2 in healthy subjects as well as colonic CD14 negative *CD1c/ CLEC10A* cluster in CD patients [31], and anti-inflammatory Tbet^+^ “cDC2A”, with the human counterpart detected in spleen and melanoma [53]. Because intestinal CD14^+^CD163^−^ MNPs do not share synovial fluid Inf Mo-DCs or circulating Inf DCs gene signature and are unable to polarize naïve T cell differentiation [54], these cells are not fulfilling DC criteria.

Rather, the colonic CD14^+^CD163^−^ MNPs display Mo-like morphology, share gene expression with monocytes (*FCAR/*CD89 and *C5AR1/*CD88), and are molecularly and functionally distinct from the CD14^bright^CD163^+^ Mɸ subset detected in inflamed IBD mucosa [31], RA synovial fluid, and tumor ascites [51]. CD14^+^CD163^−^ MNPs also share a phenotypic and molecular signature with monocyte-derived cells that increase in inflamed human small intestine, which are best characterized as CD14^dim^TREM1^+^S100A8/9^+^ cells using flow cytometry and bulk RNAseq [13]. Finally, increased frequencies of CD14^+^ cells in ileal tissue of CD patients are associated with depletion of circulating monocytes in matched blood cells; the two CD14^+^ cells express similar transcriptomic profile in blood and intestine using scRNAseq [34].

Collectively, the CD14^+^ subpopulation that accumulates in the inflamed mucosa of IBD patients can be qualified as Inf Mo-like cells, i.e., monocytes that have experienced a limited differentiation in the gut, are able to secrete pro-inflammatory cytokines, and do not display the function or gene signature of Inf Mo-DCs, Inf DCs (DC3) or Inf Mɸ (Figure 4).

### 1.4. Plasticity of Monocyte-Derived Cells in Human Inflamed Gut Mucosa

The precise nature of CD163^+^ Mɸ detected in inflamed IBD tissue also warrants further clarification. Are these cells embryo-derived, anergic, resident, or post-inflammatory Mɸ? In mice, recruited monocytes in gut mucosa were found to undergo a maturation process with a pro-inflammatory phase, followed by a second step that leads to resolution of inflammation with CD14^+^ Mɸ that have escaped inflammation and survived, actively participating in tissue repair. Notably, CD163^+^ cells have been localized around ulcers and vessels in IBD patients [55]. The sequential maturation process of CD14^+^ monocytes remains unresolved in the inflamed gut of IBD patients.

Colonic CD14^bright^CD163^+^ MNPs are best characterized by the *MAFB/SLCO2B1/STAB1/CD163L1* Mɸ gene signature in CD [31]. These tissue CD163^+^ Mɸ express several genes of late-differentiated Mɸ, a signature shared by the murine CX3CR1^hi^ Mɸ population *(MERTK/CD163L1/MRC1/ C1qa/C1qb/C1qc/Mmp12/Mmp14)* [47,56]. CD209 expression on CD14^bright^CD163^+^ Mɸ population corroborates with a high level of expression observed on the most mature Mɸ in the human jejunal mucosa at homeostasis [28]. The regulatory nature of human CD163^+^ Mɸ is highlighted by CD206 expression. Hence, CD206^+^ Mɸ are induced in IBD patients with anti-TNFα-responsiveness when compared with non-responders [57,58]. However, CD14^+^CD163^+^ Mɸ are still prone to secrete large quantities of TNFα, IL-23, along with IL-10, in inflamed CD and UC colon [31,39], not meeting the criteria for anergic Mɸ.

Do CD14^bright^CD163^+^ Mɸ arise from CD14^+^CD163^−^ Inf-Mo-like cells via a transitional CD163^dim^ population? In support of this hypothesis, CD14^+^ populations with variable levels of CD163 expression, as well as two distinct programs of Inf Mɸ, are detected within the inflammatory tissue in IBD patients [31,34]. CD14^bright^CD163^+^ Mɸ in IBD mucosa display a common gene signature with in-vitro-generated monocyte-derived Mɸ in the presence of M-CSF*(MAFB/MERTK/CD163/CD169/ CD11c/CD14/SLCO2B1/FCGRT/STAB1)* [59]. Goudot et al. showed that monocytes cultured with either M-CSF alone or a mixture of known pro- or anti-inflammatory cytokines (M-CSF, IL-4, TNF-α, and AHR) differentiate into Mɸ or DCs, whose molecular profiles are close to Inf Mɸ and Mo-DCs found in tumor ascites, respectively [60]. Although monocyte differentiation into Mɸ was suggested to be a default pathway, a recent study favoured the concept of monocyte plasticity by showing that differentiated monocytes modulate their nature with M-CSF or GM-CSF present in the environment, with cross-talk between these two pathways [61]. Tissue monocytes and Mɸ at different stages of maturation [3] likely coexist with rare embryo-derived Mɸ in inflamed tissue in vivo. Hence, CD14^+^CD163^+^ Mɸ in inflamed UC mucosa can be further subdivided into TIM4^−^CD4^-^ and TIM4^+^CD4^+^CD169^+^ subsets [39], the latter phenotype being reminiscent of embryo-derived Mɸ detected in the deep layer of jejunal wall at steady state [20].

Thus, CD14^bright^CD163^+^ Mɸ in IBD colon might be a population of post-inflammatory Mɸ, whose molecular profile indicates a repair function, while these cells simultaneously secrete pro- (TNF-α and IL-1β) and anti-inflammatory (IL-10) cytokines, thus not resembling anergic or embryo-derived Mɸ (Figure 4).

## 2. MNPs in MLNs of CD and UC patients

Mesenteric lymph nodes (MLNs) are a major site of naive T cell priming and education of memory T cells that home to gut tissue, through interactions with MNPs. In contrast with gut mucosa, few studies have examined the landscape and function of MNPs in the MLNs of IBD patients. Immunohistochemistry studies revealed the presence of conventional DC, cDC1, and cDC2 in MLNs of UC, CD, and non-IBD patients [62]. High-dimensional phenotypic mapping and transcriptional studies of MNPs, and more particularly CD14^+^ MNPs, are limited in MLNs in humans at steady state and during colitis. How these MNPs subsets are related to gut MNPs remains to be clarified. MLNs that drain the small and large intestine are anatomically and functionally segregated [63,64,65], highlighting the importance of studying MLNs at similar anatomical sites when comparing UC and CD patients (Figure 5).

Single-cell phenotypic analysis revealed that resident and migratory (CCR7^+^) CD11c^+^HLA-DR^hi^ cDCs are found in comparable numbers in both types of IBD, whereas CD11c^−^HLA-DR^dim^ plasmacytoid DCs (pDCs), which represent the main cDC population in MLNs, are detected in larger proportion in CD than UC [66]. These data apparently contradict the results reported by Granot et al. about a predominant cDC2 population in MLNs using CD11c^+^ as the parent gate [67]. A shift from HLA-DR^hi^ to HLA-DR^dim^ expression was found in CD11c^+^CD1c^+/−^ in MLN of IBD relative to healthy donors [35]. In that regard, CD14^−^CD64^dim^CD11b^+^CD36^+^CD11c^+^ that were CD1c^−^ were identified in both diseases [66]. These cells share some characteristics with circulating CD11c^dim^CD36^+^CD1c^dim^ DC3, which are identified as a separate cDC2 subset in human blood [52]. Several CD14^+^ Mɸ subsets that occupy distinct niches cohabit with DCs in MLNs. CD14^+^ cells are subdivided into CD68^+^HLA-DR^hi^ Mɸ that either express CD163, MARCO, or CD169 [66]. CD169 expression defines subcapsular (SSM) and medullary (MSM) sinus Mɸ as opposed to paracortex Mɸ that express CD68 but not CD169 [68,69]. The frequency of CD14^+^CD163^+^ MNPs is higher in UC relative to CD or non-IBD patients in MLNs and enriched in bona fide MERTK^+^MARCO^high^CD169^−^HLA-DR^hi^ Mɸ and CD11b^+^TREM1^+^HLA-DR^dim^ cells [66]. Although CD14^+^CD163^+^HLA-DR^hi^ cells in MLNs are unlikely to originate from mucosal resident CD14^bright^CD163^+^ Mɸ, two non-mutually exclusive hypotheses underscore the presence of CD14^+^CD64^+^HLA-DR^dim^ cells expressing CD163 in MLNs. The latter might be derived either from circulating monocytes directly recruited into the MLNs or from colonic CD14^+^CD163^−^ Infl Mo-like cells that acquire migratory capacities, as suggested in mice [19,70], and progressively upregulate CD163 expression upon entrance into lymph nodes. Murine Mɸ in MLNs are thought to arise from circulating monocytes in colitis. During adoptive transfer of Ly6C^hi^ monocytes in mice with colitis, the kinetics of monocyte waterfall are similar in the colon and MLNs, suggesting a direct recruitment of Ly6C^hi^ monocytes in lymph nodes rather than a migration of cells from the mucosa [18]. These massively recruited monocytes in MLN differentiate into Mɸ that contribute to colitis independent of migratory DCs. In contrast, CD MLNs are enriched in CD11b^+^CD209^−^TREM-1^+^HLA-DR^dim^CD163^−^ cells when compared to UC, which could reflect a differential and massive recruitment of mucosal CD163^−^ Inf Mo-like cells expressing TREM and *FCAR*/CD89 [66].

The transcriptomic profile of the monocyte/ Mɸ -like cells (CD14^+^CD64^+^CD163^+^) and enriched DCs (CD14**^−^**CD64**^−^**CD163**^−^**) have been compared in colonic draining MLNs of UC and CD patients using bulk RNAseq [66]. The CD14^+^CD64^+^CD163^+^ population over-expresses, relative to DCs, a mixed gene signature that best characterizes colonic CD163^+^ Mɸ (*MAFB/CSFR1/ C1QA/C1QB/ C1QC/MRC1/MAF/STAB1/SLCO2B1/FOLR2/FCGR3A/C2/VSIG4*) and Infl Mo-like cells (*CD300E/ SERPINA1/FCN1/FPR1/S100A9/SLC11A1/THBS1/ IL1RN/PLAUR/ CCRL2/OLR1*) in CD patients [31]. Notably, the CD14^+^CD64^+^CD163^+^ subset did not display the gene signature of circulating monocytes (*SELL/CLEC4D/CD48* expression), whereas they shared molecular features with in vitro human monocyte derived Mɸ [59].

Taken together, examining the immune landscape of MNPs in MLNs revealed differences between UC and CD (Figure 5). Future studies that combine scRNAseq, imaging and functional studies are warranted to define the mechanisms that regulate the preferential expansion of CD163^+^ monocyte/Mɸ-like cells in UC.

## 3. Contribution of Single-Cell Analysis in Establishment of Cell Networks and Therapeutic Implications

Single-cell technologies have contributed to accurately redefining the nature of the cells present in inflamed tissues. An additional interest lies in metadata-driven analysis that can be used to generate a comprehensive overview of cell networks potentially implicated in disease pathogenesis (Figure 6). Using scRNAseq and multicolor mass cytometry, Martin et al. showed that the inflamed ileal mucosa of a subset of CD patients displays a specific module defined by a group of cell subtypes, whose frequencies are highly correlated across patients [34]. This module, i.e., GIMATS, comprises IgG plasma cells, inflammatory Mɸ/Mo-like cells, activated cDCs, highly activated T cells, and activated stromal cells (fibroblast and ACKR1^+^ activated endothelial cells). Examining accessible gene expression data using bulk RNA sequencing analysis obtained with a large cohort of pediatric ileal CD patients (RISK), the GIMATS signature is able to predict a lower response to anti-TNFα therapy. Using an algorithm predicting specific ligand–receptor pairs’ RNA expression, the authors defined a ligand-receptor activity network in patients enriched or lacking the GIMATS module. It was consequently inferred that this cellular network is possibly implicated in vivo in the inflammatory process and resistance to therapy [34]. Similarly, using scRNAseq of inflamed mucosa in UC patients, Smillie et al. reported an increase in several cell types, notably inflammatory monocytes, cDC2, CD8^+^IL-17^+^ T cells, TNF-α^+^ T regulatory cells, IgG plasma cells, follicular cells, microfold M cells, BEST4^+^ enterocytes, and IL13RA2^+^IL-11^+^ inflammation-associated fibroblasts (IAF) [45]. Comparing their data with a meta-analysis of bulk expression data from responders and non-responders to anti-TNFα therapy, the drug resistance signature was associated with enrichment in inflammatory monocytes, cDC2, and IAF [45]. Since IAF was enriched in patients resistant to anti-TNFα therapies, the authors concluded that IAF, which expresses the oncostatin M (OSM) receptor, may be implicated in the OSM-mediated resistance to anti-TNFα reported by West et al. [71]. In an independent cohort of UC patients, a disease core gene signature identified in rectal biopsies at baseline can predict corticoid response and, overlap with gene signature previously associated with anti-TNFα and α4β7 response [72].

## 4. Conclusion and Future Perspectives

Single-cell technologies are revolutionizing tools that provide an unprecedented view of nonimmune and immune cell networks implicated in health and disease. These technologies might help to decipher the individual contribution of the different players potentially implicated in the pathophysiology of complex diseases like IBD. They provide the potential to predict response to therapy and offer a personalized medicine approach to a heterogeneous group of patients with IBD. Advances in the development of novel therapeutic approaches, prediction of therapeutic responses, optimization of individualized medicine, and ultimately disease prevention will highlight the relevance of performing single-cell analysis for understanding the molecular basis of IBD. Before using this information in clinics, we should be cautious to not draw early conclusions of data mainly driven by powerful algorithms. Hence, mathematical modeling that proposes novel hypotheses (bottom-up) will require functional validation and therefore should remain complementary to hypothesis-driven models that lead to molecular data validation (top-down) to unravel IBD pathogenesis and offer translational therapeutic perspectives [73] (Figure 6).

## Figures and Tables

**Figure 1 cells-09-00813-f001:**
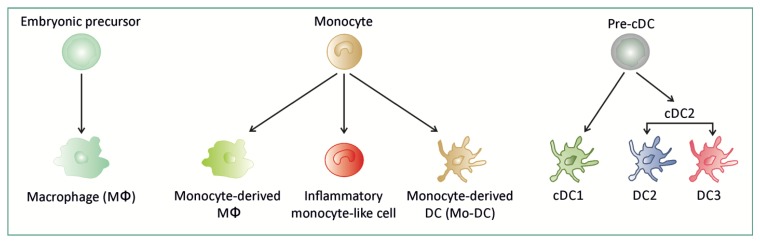
Ontogeny of macrophages, monocyte-derived cells and dendritic cells. Macrophage (Mɸ) can be derived from embryonic precursor or monocyte. Monocyte can differentiate into Mɸ, dendritic cell (Mo-DC) or tissue inflammatory monocyte-like cell. Conventional dendritic cells (cDC1 and cDC2 that is further subdivided into DC2 and DC3 subsets) originate from a dedicated precursor (pre-cDC).

**Figure 2 cells-09-00813-f002:**
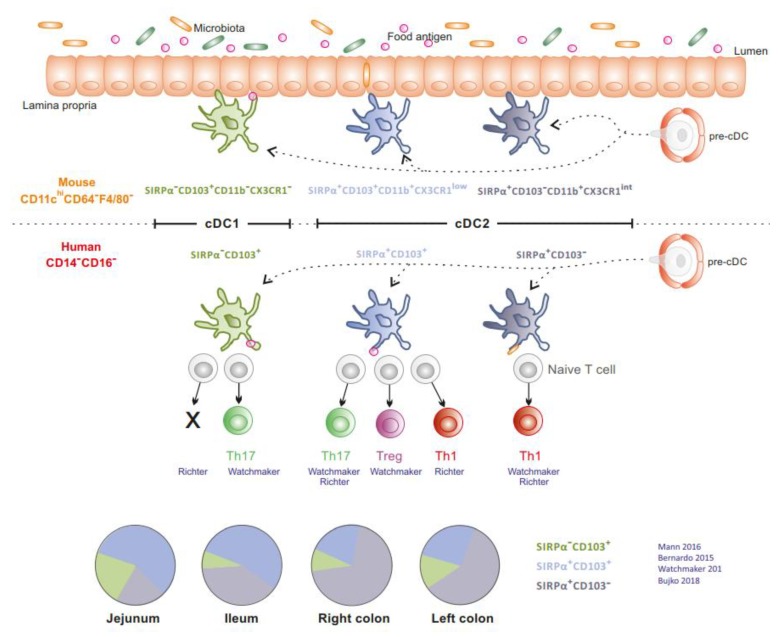
Distribution and function of intestinal conventional DCs. SIRPα^−^CD103^+^ cDC1, SIRPα^+^CD103^+^ cDC2, SIRPα^+^CD103^−^ cDC2 are relatively conserved between mice and humans. Human jejunal cDCs prime allogenic naive CD4^+^ T cells and promote differential T cell responses. The proportion of human cDC subsets according to their gut location is depicted in the pie charts (Literature cited in blue).

**Figure 3 cells-09-00813-f003:**
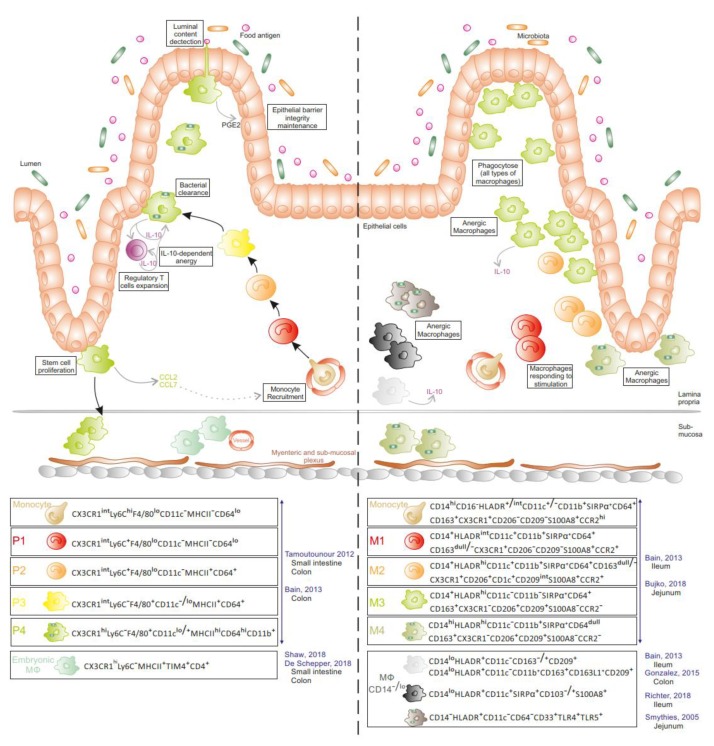
Intestinal monocyte/macrophage populations at steady state in mice and humans. In mice (left panel), the majority of Mɸ derive from circulating monocytes. Once recruited in tissue, monocytes undergo a maturation process into anergic Mɸ (P1 to P4), producing and responding to IL-10 and ensuring diverse functions in the lamina propria. Embryonic Mɸ are located in the submucosa. In humans (right panel), monocyte derived-Mɸ (M1 to M4), the potential counterparts of murine mononuclear phagocytes (MNPs) and several anergic Mɸ subsets have been reported in the lamina propria. (Literature cited in blue along with gut location).

**Figure 4 cells-09-00813-f004:**
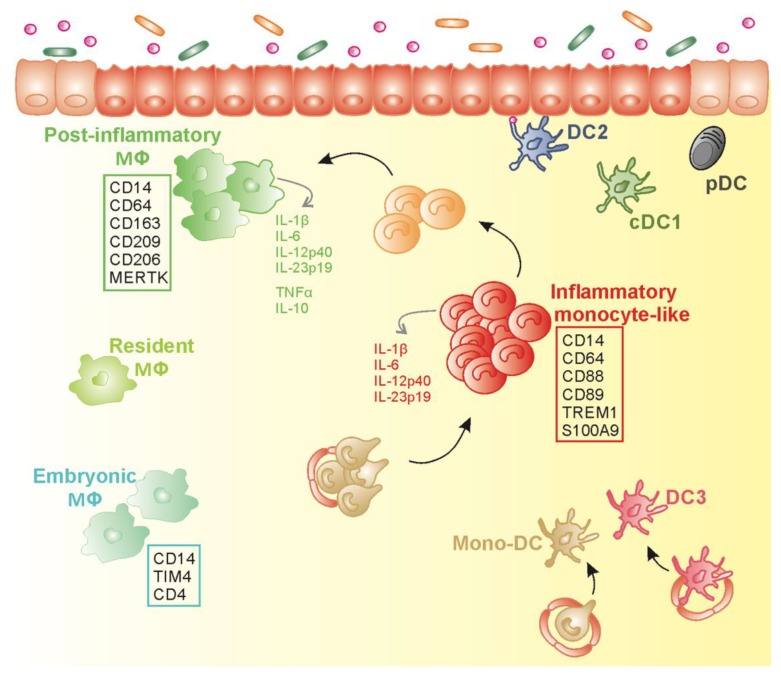
Proposed schematic model for mononuclear phagocytes diversity in inflamed colon of inflammatory bowel disease (IBD) patients. In inflamed IBD gut mucosa, the accumulation of HLADR^dim^CD14^+^CD163^−^CD89^+^TREM^+^ inflammatory monocyte-like subset (Inf Mo-like) (in red) secreting pro-inflammatory cytokines, could result from the increase recruitment of circulating CD14^hi^ monocytes (in gold) that differentiate into Inf Mo-like cells in concert with the potential arrest in the maturation program towards HLADR^hi^CD14^hi^CD209^+^MERTK^+^ post-inflammatory Mɸ (in green) that likely contribute to tissue repair. Transitioning cells (in orange) are generated during this maturation process. Post-inflammatory Mɸ coexist with resident Mɸ (in yellow–green) that represent the predominant Mɸ population at steady state. Mɸ expressing TIM-4^+^ and CD4^+^ (in mint green), like embryonic Mɸ reported in mice, have been identified in the inflamed colon of IBD patients. Besides Inf Mo-like cells and Mɸ, conventional dendritic cells that include cDC1 (in khaki), DC2 (in blue), and plasmacytoid DC (in black) are seeded in the inflamed mucosa. Inflammatory monocyte-derived DC (in gold) and inflamed DC3 (in dark pink) may infiltrate inflamed lamina propria in IBD patients.

**Figure 5 cells-09-00813-f005:**
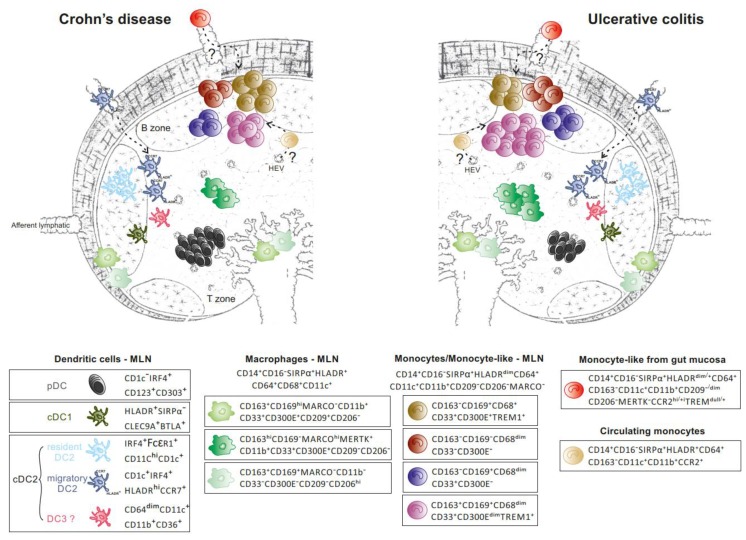
MNPs in human mesenteric lymph nodes in Crohn’s disease and ulcerative colitis. Rare SIRPα^−^ cDC1 (in khaki) and four CD14^−^CD64^−^CD163^−^ DCs subsets: *(1)* pDCs (in black), the major DC subset found at higher frequency in CD compared to UC; *(2)* resident CD11c^hi^CD1c^+^CD33^+^ DC2 (in light blue); *(3)* rare migratory HLADR^hi^CDR7^+^ DC2 (in dark blue) and *(4)* CD163^int^CD11b^+^CD36^+^CD1c^−^ cDCs (in pink), reported in similar proportion in CD and UC. A higher proportion of HLADR^dim^CD68^dim^CD169^+^ monocyte-like cells (in purple) and HLADR^hi^CD68^+^MERTK^+^CD169^−^ Mɸ (in dark green) contributes to increased frequency of CD14^+^CD64^+^CD163^+^ cells in UC compared to CD. The former (purple) could derive from circulating monocytes (in gold) directly entering MLN, or mucosal monocyte-like cells (in red) that have acquired CD163 and migratory capacities. CD169^+^ Mɸ (2 subsets depicted in light green) display a sinusoidal-like Mɸ phenotype. HLADR^dim^CD14^+^CD64^+^CD163^−^ monocyte/monocyte-like cells (brown, burgundy and navy blue).

**Figure 6 cells-09-00813-f006:**
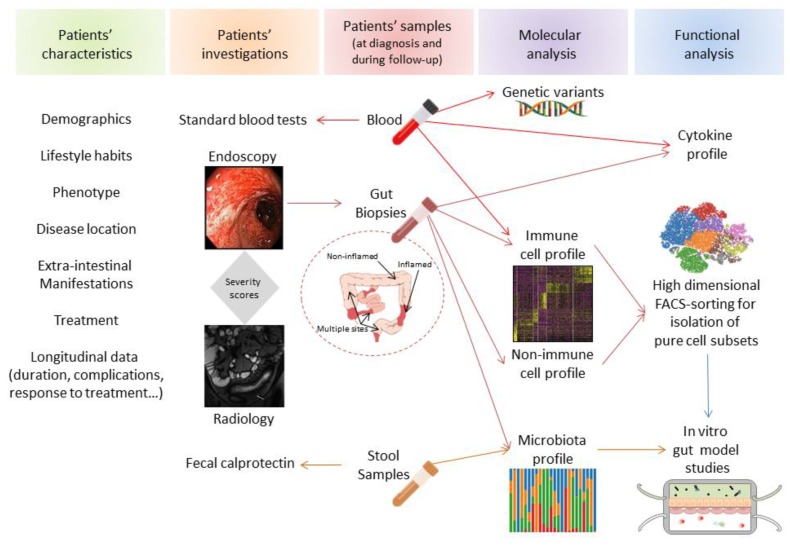
Complementary axis for personalized medicine in IBD. Different translation axis of data (patients characteristics, investigation techniques combined with molecular and functional analysis of biological samples) can potentially establish a comprehensive score that might ultimately offer clinicians therapeutic perspectives for individual IBD patient.

**Table 1 cells-09-00813-t001:** Gene and protein expression on intestinal monocytes, inflammatory monocyte-like and macrophages (function as referenced at https://www.ncbi.nlm.nih.gov/gene).

Gene	Protein	Name: Function	Expressed on
*CD14*	CD14	Co-receptor for bacterial lipopolysaccharide	Monocyte/ Infl. Monocyte/ Mɸ
*FCGR1A*	CD64	Fc Fragment of IgG Receptor Ia	Monocyte/ Infl. Monocyte/ Mɸ
*SIRPA*	SIRPalpha/CD172a	Signal regulatory protein alpha: receptor-type transmembrane glycoproteins known to be involved in the negative regulation of receptor tyrosine kinase-coupled signaling processes	Monocyte/ Infl. Monocyte/ Mɸ
*ITGAX*	CD11c	Integrin subunit alpha X: encodes the integrin alpha X chain protein	Monocyte/ Infl. Monocyte/ Mɸ
*ITGAM*	CD11b	Integrin subunit alpha M: encodes the integrin alpha M chain	Monocyte/ Infl. Monocyte/ Mɸ
*CCR2*	CCR2	C-C motif chemokine receptor 2: receptor for monocyte chemoattractant protein-1, a chemokine which specifically mediates monocyte chemotaxis	Monocyte/ Infl. Monocyte
*TREM1*	TREM1	Triggering receptor expressed on myeloid cells 1: encodes a receptor belonging to the Ig superfamily that is expressed on myeloid cells	Monocyte/ Infl. Monocyte
*C5AR1*	C5AR1/CD88	complement C5a receptor 1	Monocyte/ Infl. Monocyte
*FCAR*	CD89	Fc Fragment of IgA Receptor	Monocyte/ Infl. Monocyte
*FPR1*	FPR1	Formyl Peptide Receptor: G protein-coupled receptor of mammalian phagocytic cells	Monocyte/ Infl. Monocyte
*S100A9*	S100A9	S100 Calcium Binding Protein A9 (Calprotectin)	Monocyte/ Infl. Monocyte
*CD300E*	CD300E	Probably acts as an activating receptor	Monocyte/ Infl. Monocyte
*SLC11A1*	SLC11A1	Solute carrier family 11 member 1: member of the solute carrier family 11 (proton-coupled divalent metal ion transporters) family encoding a multi-pass membrane protein	Monocyte/ Infl. Monocyte
*VCAN*	VCAN	Versican: large chondroitin sulfate proteoglycan, major component of the extracellular matrix. Involved in cell adhesion, proliferation, migration and angiogenesis; plays a central role in tissue morphogenesis and maintenance	Monocyte/ Infl. Monocyte
*CD68*	CD68	Member of the lysosomal/endosomal-associated membrane glycoprotein (LAMP) family and scavenger family	Mɸ
*CD163*	CD163	Member of the scavenger receptor cysteine rich (SRCR) superfamily	Mɸ
*CD163L1*	CD163L1	Scavenger receptor cysteine-rich type 1 protein M160	Mɸ
*MERTK*	MERTK	MER proto-oncogene, tyrosine kinase: member of the MER/AXL/TYRO3 receptor kinase family	Mɸ
*CD209*	CD209	Encoded a transmembrane receptor involved in the innate immune system and recognizes numerous evolutionarily divergent pathogens	Mɸ
*MRC1*	CD206	Mannose receptor C-type 1: membrane receptor that mediates the endocytosis of glycoproteins by macrophages	Mɸ
*MAFB*	MAFB	Basic leucine zipper (bZIP) transcription factor	Mɸ
*STAB1*	STAB1	Stabilin-1: encodes a large, transmembrane receptor protein which may function in angiogenesis, lymphocyte homing, cell adhesion, or receptor scavenging	Mɸ
*SLCO2B1*	SLCO2B1	Solute carrier organic anion transporter family member 2B1: member of the organic anion-transporting polypeptide family of membrane proteins	Mɸ
*C1QA*	C1QA	Complement C1q A chain: associates with C1r and C1s to yield the first component of the serum complement system	Mɸ
*C1QB*	C1QB	Complement C1q B chain: associates with C1r and C1s to yield the first component of the serum complement system	Mɸ
*C1Qc*	C1Qc	Complement C1q C chain: associates with C1r and C1s to yield the first component of the serum complement system	Mɸ
*MMP12*	MMP12	Matrix metallopeptidase 12: encodes a member of the peptidase M10 family of matrix metalloproteinases	Mɸ
*MMP14*	MMP14	Matrix metallopeptidase 14: encodes a member of the peptidase M10 family of matrix metalloproteinases	Mɸ
*TIMD4*	TIM4	T cell immunoglobulin and mucin domain containing 4	Embryonic Mɸ

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
