# Peer review of "Single-Cell Protein and RNA Expression Analysis of Mononuclear Phagocytes in Intestinal Mucosa and Mesenteric Lymph Nodes of Ulcerative Colitis and Crohn’s Disease Patients"

_cells, 2020, doi:10.3390/cells9040813_

Round 1

Reviewer 1 Report

A few comments:

  1. Abstract should be expanded that includes some monumental findings in a few extra sentences, as long as it is within the words limit;
  2. Should include more references/citation that impact the field.

Author Response

Referee #1 (underlined in yellow)

  1. We expanded the abstract (lines 14-16 and 19-20).
  2. We have included 3 recent references in lines 64-66 (#17,2020), in line 298 (#65,2019), in lines 378-380(#72,2019).

Reviewer 2 Report

The review article “Single-Cell Protein and RNA Expression Analysis of Mononuclear Phagocytes in Intestinal Mucosa and Mesenteric Lymph Nodes of Ulcerative Colitis and Crohn’s Disease Patients” describes landscape of mononuclear phagocytes (MNP) in non-lymphoid and lymphoid tissues of IBD patients. The review nicely outlines unique clusters identified using scRNA sequencing/multicolor flow cytometry. The authors conclude that single- cell technologies could provide clue to pathophysiology and therapeutics of complex diseases such as IBD, via identification of novel cellular networks. Description of novel MNP subtypes in intestinal mucosa is interesting for the readers; however, some parts require better clarity of presentation.

  1. In many places, authors try to convey a lot of information in complex sentences, which are difficult to comprehend. For clarity, very long sentences should be simplified.
  2. A table summarizing the MNP phenotype, subsets and location (small vs large intestine) along with known functions would be helpful for readers.
  3. The section on MNPs in mesenteric lymph node in IBD adds novelty to the review. A pictorial /tabular representation would enhance this section.
  4. Authors have compared MNPs in healthy and IBD patients or murine colitis. The gene signatures and surface markers could also be summarized in a table or figure (in both human and mouse).

Author Response

Referee #2(underlined in yellow)

  1. Long sentences have been shortened ( ex: line 70, 121, 221).
  2. We have inserted a new Fig 2  (Distribution and function of  intestinal conventional DCs, lines 45-60) and  a new Fig 3 (Intestinal monocyte/macrophage populations at steady state in mice and humans, lines 107-115)
  3. We have inserted a new Fig 5 (MNPs in human Mesenteric Lymph Nodes in Crohn’s disease and Ulcerative colitis , lines 340-352) 
  4. We did not insert a new fig describing MNPs in murine colitis. We believe that such a figure would be redundant with Fig 3 (left panel) since it would just illustrate the arrest in the maturation process as stated “Although no unique experimental animal model of IBD entirely replicates human disease, during colitis, the sequential maturation process of monocytes into anti-inflammatory Mɸ (P4) is interrupted. This arrest of maturation process, in combination with massive recruitment of circulating monocytes, promotes the accumulation of CX3CR1int (P1, P2, and P3) subsets that secrete pro-inflammatory cytokines in inflamed gut” (lines 121-123).

Reviewer 3 Report

This review is well written regarding single-cell analysis of protein and RNA expression of mononuclear phagocytes in IBD. In particular, the analysis with this method might be useful to see the responsibility to anti-TNFα therapy in IBD patients (chapter 3).

I would suggest a few minor points as follows.

  1. A short summary of abbreviations of various kinds of markers and marker-positive macrophages would be more understandable for readers.

  1. It would be clearer to describe what kinds of experimental models were used for analysis in murine intestinal mucosa during inflammation (Chapter 1.2).

  1. Prostaglandin E-major urinary metabolite (Patients’ investigations)* from the urine (Patients’ samples) could be added at the bottom of Figure 3.

*Arai Y et al. Prostaglandin E-major urinary metabolite as a reliable surrogate marker for mucosal inflammation in ulcerative colitis. Inflamm Bowel Dis. 2014;20:1208-16.

Hagiwara SI et al. Prostaglandin E-major urinary metabolite as a biomarker for pediatric ulcerative colitis activity. J Pediatr Gastroenterol Nutr. 2017;64:955-61.

Author Response

Referee #3(underlined in yellow)

  1. A summary of abbreviations has been included in Table 1(lines 203-206)
  2. Experimental models used in references cited : Dextran Sodium Sulfate (DSS)-induced and T cell-mediated colitis” in lines 118 and 215.
  3. We are hesitating to insert Prostaglandin E-major urinary metabolite in the revised Fig 6 (previous Fig 3) since this test is not yet included/ recognized in “ Standard care routine test”. We might include it upon further request from the referee.

Round 2

Reviewer 2 Report

The changes incorporated by authors are satisfactory and have improved the manuscript.